# Acoustic Monitoring of Black-Tufted Marmosets in a Tropical Forest Disturbed by Mining Noise

**DOI:** 10.3390/ani13030352

**Published:** 2023-01-19

**Authors:** Esther Bittencourt, Angélica da Silva Vasconcellos, Renata S. Sousa-Lima, Robert John Young, Marina Henriques Lage Duarte

**Affiliations:** 1Laboratory of Bioacoustics, Museum of Natural Sciences, Pontifical Catholic University of Minas Gerais, Belo Horizonte 30535-901, Brazil; 2Post Graduate Program in Vertebrate Biology, Pontifical Catholic University of Minas Gerais, Belo Horizonte 30535-901, Brazil; 3Laboratory of Bioacoustics (LaB) and EcoAcoustic Research Hub (EAR Hub), Department of Physiology & Behaviour, Federal University of Rio Grande do Norte, Natal 59078-970, Brazil; 4School of Science, Engineering and Environment, University of Salford, Peel Building, Manchester M5 4WT, UK

**Keywords:** animal communication, mining activity, noise pollution, primates

## Abstract

**Simple Summary:**

Communication is one way that animals use to obtain and defend resources, escape predators and attract sexual partners. However, this process can be disrupted by anthropogenic noise, which often differs from natural sounds in frequency, duration and intensity. This study aimed to understand whether, and how, calls emitted by black-tufted marmosets (*Callithrix penicillata*) are affected by mining noise. We compared ambient noise and the acoustic parameters of the contact calls of these animals in two study areas, one near and one far from the Brucutu Mine, Minas Gerais, Brazil. We found background noise to be higher in the area near the mine, and marmoset vocalizations more frequent, compared to the far area. Calls emitted in the near area also differed in spectral parameters from the far area, which suggests an effort by the animals to adapt their vocal activity to a noisier environment. Our results indicate that mining noise may affect the acoustic communication of black-tufted marmosets. These results may be taken as a starting point for establishing public policies to promote preventive and/or mitigative measures to protect wildlife around sites of mining activity. Moreover, measures to regulate any noisy activities in relation to wild animals are pressing since these are lacking in Brazil.

**Abstract:**

All habitats have noise, but anthropogenic sounds often differ from natural sounds in terms of frequency, duration and intensity, and therefore may disrupt animal vocal communication. This study aimed to investigate whether vocalizations emitted by black-tufted marmosets (*Callithrix penicillata*) were affected by the noise produced by mining activity. Through passive acoustic monitoring, we compared the noise levels and acoustic parameters of the contact calls of marmosets living in two study areas (with two sampling points within each area)—one near and one far from an opencast mine in Brazil. The near area had higher anthropogenic background noise levels and the marmosets showed greater calling activity compared to the far area. Calls in the near area had significantly lower minimum, maximum and peak frequencies and higher average power density and bandwidth than those in the far area. Our results indicate that the mining noise affected marmoset vocal communication and may be causing the animals to adjust their acoustic communication patterns to increase the efficiency of signal propagation. Given that vocalizations are an important part of social interactions in this species, concerns arise about the potential negative impact of mining noise on marmosets exposed to this human activity.

## 1. Introduction

An increasing number of studies have shown that anthropogenic noise negatively affects communication in both human and nonhuman animals [1]. These negative effects are not limited to communication, and can also affect the health of humans and animals; for example, noise may cause sleep disturbances, contribute to cardiovascular disease and even lead to the development of mental illnesses, among other problems [2]. Recently, an increase in the occurrence of anthropogenic noise throughout the planet has been recognized—and not just in areas of human activities such as mining [3]. Documented effects of noise on wildlife include modulations in acoustic communication [4,5,6], as well as other behavioral changes, such as decreased time spent foraging and sleeping [5], and physiological responses, such as increased secretion of hormones indicative of stress [5]. In addition, ecological impacts, such as changes in species distribution, richness and composition have been documented, e.g., [7,8]. Noise is cited by some authors as significantly contributing to biodiversity decline in all ecosystems in the world [9].

A wide range of taxa, from insects to mammals, use sound to exchange vital information that mediates survival (territorial defense, food begging, coordination of group foraging) and reproduction (mate attraction, resource holding display) [9]. Sound emission represents an important means of communication with conspecifics for most vertebrate species [10,11]. Many species that inhabit tropical forests, especially primates, use vocalizations as a means of communicating over long distances [12]. However, this process can be negatively affected by anthropogenic noise through masking. Masking occurs when a sound is emitted at the same time as animal vocalizations, and at the same frequency, with enough energy that it affects the perception of the signal of interest [7]. The degree to which an acoustic signal is masked by noise depends on the degree of spectral and temporal overlap between signal and noise, as well as the auditory sensitivity of the receiver [9].

Mining is an important economic activity worldwide, but especially in the Quadrilátero Ferrífero (Iron Quadrangle), located in the state of Minas Gerais, Brazil [13]. Due to its nature, opencast mining inevitably leads to serious habitat loss with several significant impacts on the environment [14]. Among the various impacts generated by mining, the production of noise from truck traffic, explosions and machinery operation has the potential to mask animal sounds and affect their long-term behavior and distribution [7]. Despite the existence of research that shows the impacts of mining noise on wildlife, there are still no laws in Brazil that regulate and require the monitoring of this anthropogenic noise on animals, or that establish a maximum level of noise pollution in areas not inhabited by humans [15].

*Callithrix penicillata* is a primate species endemic to Brazil that occurs mainly in the Cerrado but can also be found in the Atlantic Forest and, seasonally, in the dry forest, “Caatinga”, or even inhabiting urban green areas [16]. These animals have arboreal habits and live in family groups [17]. Primates of this genus have thirteen described types of vocalizations [18], which, according to Epple (1968) [19], can be classified into three major groups according to their social function: “contact calls”, “mobbing calls” and “alert calls”. Among contact calls, we highlight the *phee*-type vocalization. This type of call is known as a long-distance call, and is more frequently emitted by adults than by juveniles [19]. The *phee* call is normally used to establish contact among conspecifics, to reunite group members at dusk to rest, when an individual is isolated or when close visual contact is maintained with another animal [10,18,19]. Some studies show that marmosets can alter their sounds to avoid masking, by modulating some temporal and spectral parameters of their vocalizations [6,20,21].

Understanding the effects of human-induced environmental changes, such as noise pollution, is important for designing targeted conservation efforts [8]. This research aimed to analyze the effects of mining noise on the acoustic communication of *Callithrix penicillata*. We hypothesized that there will be a difference in ambient sound levels between areas near and far from mines and, consequently, that animals in these different areas will change contact call emission rates and the spectral and temporal parameters of these contact calls.

## 2. Materials and Methods

### 2.1. Study Area

The study was carried out at the Estação Ambiental de Peti (Figure 1), located in an area of transition between the Cerrado and the Atlantic Forest. The area is crossed by the Santa Bárbara River, which belongs to the Doce River Basin in São Gonçalo do Rio Abaixo, state of Minas Gerais, Brazil. The study area has a predominantly tropical and warm temperate climate, with a dry period in winter. Rainfall patterns are strongly seasonal with 110 mm mean rainfall, mean temperature of 23.9 °C and 61.9% humidity in the rainy season (October to March). During the dry season, from April to September, the region is characterized by 13 mm mean rainfall, mean temperature of 18 °C and 58.1% humidity [6]. Thirty species of amphibians, eighteen species of reptiles [22], 231 species of birds [23] and 30 species of mammals [24] are known to inhabit the Peti reserve.

The Estação Ambiental de Peti (Peti reserve) is contiguous with the Brucutu Mine, one of the largest iron mines in the world [25]. Located in the Quadrilátero Ferrífero, this region is characterized by the highest concentration of operating opencast mines in the world, and includes the Metropolitan Region of the state capital, Belo Horizonte [7].

### 2.2. Data Collection

Data were collected using passive acoustic monitoring methods to record soundscapes where two groups of black-tufted marmosets (*Callithrix penicillata*) inhabit. Four autonomous recorders (Song Meter SM2; Wildlife Acoustics) were installed, two in a forest fragment 100 m away from the Brucutu Mine (near area), and two in a forest fragment 2500 m away from the mine (far area). The distance between the two areas was approximately 2300 m. Geographic barriers between the sampling areas (some roads and a river) probably isolated the individuals from these two groups, preventing them from interacting. The size of the home range of the species is small enough that these groups of animals (inhabiting “near” and “far” areas) would not come into auditory contact of each other [26]. The two chosen forest fragments were matched in terms of size, floristic composition and habitat structure (unpublished data obtained through a fieldwork pilot).

The recorders were programmed to record the soundscape between the periods of 5:00 a.m. and 6:00 p.m. (marmosets are a strictly diurnal species and this collection period coincides with their activity cycle [27]) for seven days, every two months, between October 2012 and August 2013. In total, six recording sessions were analyzed, summing 1092 h (two areas X six recording sessions X 13 h for seven consecutive days). Each recorder was fixed to a tree 1.5 m above ground, leaving the two side microphones unobstructed for recording. Recordings were browsed manually to detect *phee*-type calls and noise events generated by truck passes. Six spectral parameters were used to characterize marmoset *phee*-type calls in the two sampling areas: call duration, maximum frequency, minimum frequency, peak frequency, mean power and bandwidth. These parameters were calculated through spectrograms generated in Raven Pro 1.6 software (Cornell Lab of Ornithology; Figure 2). To quantify the disturbance from truck traffic, we counted all instances of noise generated by passing trucks from 5:00 a.m. to 6:00 p.m. on the road in front of the site near to the mine. Hourly counts were performed by audio and visual identification of the trucks’ broadband pattern in spectrograms (fast Fourier transform 1024 points, filter bandwidth 135 Hz; frequency resolution, 93.8 Hz; grid time resolution 2.13 ms) generated in Raven Pro 1.6 (Cornell Lab of Ornithology, Ithaca, NY, USA). In addition, the temporal distribution and emission rate of vocalizations and truck noise events were analyzed, using the Oriana software (Kovach Computing Services, Anglesey, UK).

In order to verify differences in sound levels (measured in decibels—dB (A)) in the areas near and far from the mine, measurements of noise pollution levels were also made using sound level meter frequency analyzers B&K2270 (Virum, Denmark). In order to avoid bias in the measured sound pressure levels, we excluded all recordings in which we verified high received levels of animal sounds by using BZ 5503 software. We conducted one measure by hour, from 05:00 a.m. to 06:00 p.m., with 20 min duration each one, in each sampling point per sampling area. The measures were conducted in both areas simultaneously, once every two months during data collection. For each noise measurement, we calculated the equivalent continuous level (Leq; time average level of sound), which gives an overall indication of the level of exposure to sound in an environment [6,27].

### 2.3. Statistical Analyses

Acoustic parameters of *phee*-type calls (duration, minimum frequency, maximum frequency, peak frequency, average power and bandwidth) and noise levels of both areas (near and far, the noise decibel levels were averaged per hour) were compared through generalized linear models (GLMs), with Poisson distribution. We used the “lme4” [28], “MASS” [29] and “car” [30] packages to fit the models in R statistical software, version 3.5.2 (R [31]). All results were analyzed based on α ≤ 0.05 statistical significance.

## 3. Results

Noise levels ranged from Leq 33.3 dB (A) to Leq 38.8 dB (A) in the far area, and from Leq 38.7 dB (A) to Leq 70.9 dB (A) in the near area. Values obtained from the comparison of noise and acoustic parameters of *phee*-type vocalizations between the two study areas are shown in Table 1. Both noise level and vocalization emission rates were significantly higher in the near area than in the far area. Spectral (call duration, minimum frequency, maximum frequency, peak frequency, average power and bandwidth) and temporal parameters of the calls also differed significantly between the two groups of marmosets inhabiting the near area and the far area.

*Phee*-type vocalizations in the noisy (near) area had a significant longer duration and lower minimum, maximum and peak frequencies. In contrast, the average power and bandwidth of the *phee* calls were significantly higher in the near area compared to the far area.

Analysis of the temporal distribution of *phee* vocalizations and noise events generated by passing trucks (Figure 3) shows that marmosets vocalized mostly in the period between 6:00 am and 12:00 pm, at both study areas (near and far). This period is also characterized by a higher level of noise from truck traffic at the near area.

## 4. Discussion

Our results showed that the levels of noise pollution differed between the study areas (near and far from the Brucutu Mine), even though the areas were only 2300 m apart, with noise being louder in the near area. Quantitative analyses indicated that marmosets emitted contact calls more frequently in the near area compared to the far area. The acoustic parameters of *phee* vocalizations also showed differences between study areas, with lower maximum, minimum and peak frequencies in the noisier (near) area, as well as greater intensity and duration of contact calls where noise was more prevalent.

The increase in the contact calling rate in the near/noisier area can be interpreted as an attempt by the animals to maintain group cohesion by increasing vocalization efficiency through call repetition. Likewise, the increase in call duration follows the same rationale, suggesting that the animals made longer calls where noise was more prevalent [4,6]. These results contrast with a study on black-fronted titi monkeys (*Callicebus nigrifrons*), which made significantly fewer loud calls closer to a mine [6]. This contrasting evidence of the effect of noise on these two primate species could be attributed to the nature of these different calls. Titi monkey loud calls are probably more energy costly than the *phee*-type contact calls emitted by the marmosets. Therefore, where noise is more prevalent, the use of costly calls that are likely masked is not as cost-effective, but otherwise, for marmosets, contact calling more often is efficient in maintaining group cohesion at a comparatively lower cost.

Nonetheless, the greater average power recorded for the vocalizations of marmosets near the mine is suggestive of the occurrence of the Lombard Effect [32]; that is, an increase in signal amplitude in response to masking. The Lombard Effect has already been observed in marmosets, birds and bats, which emit longer and more powerful syllables when exposed to increased noise intensity [32,33,34,35], inasmuch an increased cost of communication is likely for marmosets within the area near the mine.

Some studies have shown that animals use a range of vocal adjustments to minimize the immediate impact of noise on their communication [6]. For example, black-tufted marmosets (*Callithrix penicillata*) may increase the signal amplitude of their calls and the duration of notes in response to increased noise levels [32]. These vocal adjustments used by the animals can increase energetic costs due to the energy involved in producing louder or more frequent vocal signals. Furthermore, vocal adjustments do not guarantee successful communication and animals may lose contact with offspring, mating opportunities and perception of predator alarm calls from group members [4].

Our results contrast with the most common findings for birds [36] and cetaceans [37], for which minimum frequencies are higher in the presence of noise, which minimizes the overlap between calls and noise. Increasing the minimum frequency seems to be the obvious behavioral alteration used by animals to compensate for background noise since anthropogenic noise usually occupies low frequencies in acoustic space. Our results showed the opposite but were similar to those found by Santos et al. [34], who compared *phee*-type vocalizations of black-tufted marmosets living in an urban park and in a natural area far from noise pollution. That study [34] also found marmosets emitting vocalizations with lower minimum frequencies in the noisier study area. The similarity between the present results and those of Santos et al. [34], even when using different recording equipment and in different study areas, suggests that the noise compensation strategy—more frequent vocalizations with lower maximum, minimum and peak frequencies—is a typical strategy of the black-tufted marmoset to surpass noise. Human disturbance can change the way in which acoustic signals are used by animals to communicate [38] since higher frequencies are less efficiently propagated over longer distances than lower frequencies [39]. Therefore, we agree with Santos et al. [34] in their suggestion that marmosets decrease the frequency of contact calls to improve their range of communication since *phee* calls are used to maintain contact among distant group members.

The differences observed between the calls recorded in the areas near and far from the mine support our hypothesis that the animals would alter the rate and spectral parameters of their calls in the noisier (near) area. Communication between individuals across a complex physical environment, such as areas of native vegetation, is only successful when signals are propagated effectively and transmitted efficiently in the presence of ambient noise. Increased anthropogenic noise has created environmental pressures that constrain communication in new ways and can negatively impact animal populations [40]. Acoustic communication requires vocalization effort, defined by Miksis-Olds and Tyack [41] as the expenditure of energy to produce a call as a function of vocalization rate, duration and frequency. In this sense, by increasing vocalization frequency and duration, animals may be increasing their energy expenditure [4], which may have a negative impact on their reproductive success [42]. Finally, noisy environments may be avoided by the animals due to their potential to promote high levels of stress and decrease foraging opportunities [28,43], thus, potentially reducing an environment’s carrying capacity for certain species.

By altering animal communication, through impacts on the environment, humans have the potential to drive changes in the population dynamics of several species, as well as in the interspecific interactions and ecosystem functions with which these species are involved [44]. Our study showed evidence of vocal compensation in the communication system of a wild primate species in a noisy environment [32]. However, our study has some limitations; for example, the small number of sampling sites. Data from more study areas—at different distances from the mine—would contribute to a broader understanding of the effects recorded. However, we had some logistical limitations in our study that precluded us from collecting more data: (1) the mine is private, which makes the process to obtain access to these areas very bureaucratic; (2) finding secure areas to install the recording equipment—which is very expensive—is not easy; and (3) this study requires an affected and a control area with known groups of animals, which limits the possibilities to find an eligible area. Nevertheless, the data presented here fill a gap in science since most studies on the impact of noise on animals have focused on birds [20]. Our results, therefore, contribute to the growing body of scientific evidence showing the need to monitor and regulate noise pollution around mining areas close to natural areas. The results of this research will, therefore, contribute to the creation of laws to regulate, mitigate and control the impacts of anthropogenic noise pollution on wildlife.

## 5. Conclusions

In this work, we recorded differences in the vocalization rate and spectral parameters of black-tufted marmoset contact calls between an area near a mine and an area far from it. The occurrence of more frequent vocalizations, with greater duration and intensity in areas near the mine, and with lower minimum, maximum and peak frequencies, suggest that mining noise has an impact on the communication of these animals. These changes can increase the cost of communication for these primates, who have to increase their energy expenditure for sound production in an attempt to maintain communication efficiency. Our results have the potential to contribute to the elaboration of laws regulating the levels of noise that can be generated around natural areas.

## Figures and Tables

**Figure 1 animals-13-00352-f001:**
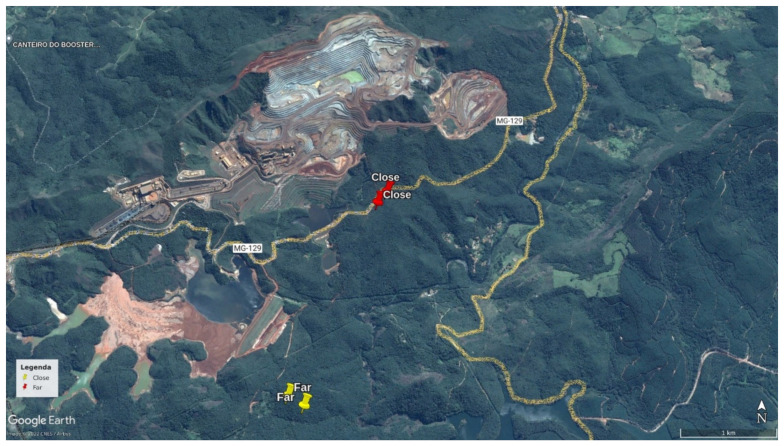
Position of the two acoustic sensors within the study areas (one near and one far from the Brucutu Mine), both within the Peti reserve, Minas Gerais, Brazil.

**Figure 2 animals-13-00352-f002:**
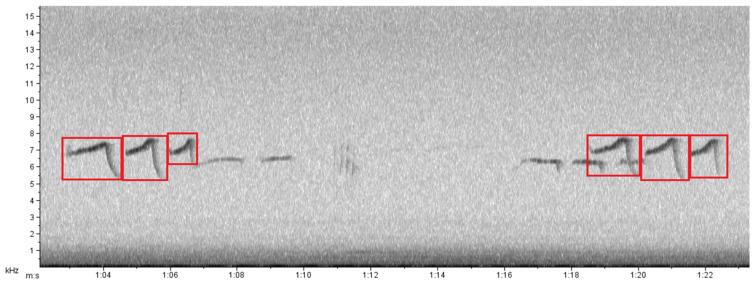
Spectrogram generated in Raven Pro 1.6 software with *phee*-type vocalizations (in red squares) of the species *Callithrix penicillata*, recorded in the morning at the Peti reserve, Minas Gerais, Brazil.

**Figure 3 animals-13-00352-f003:**
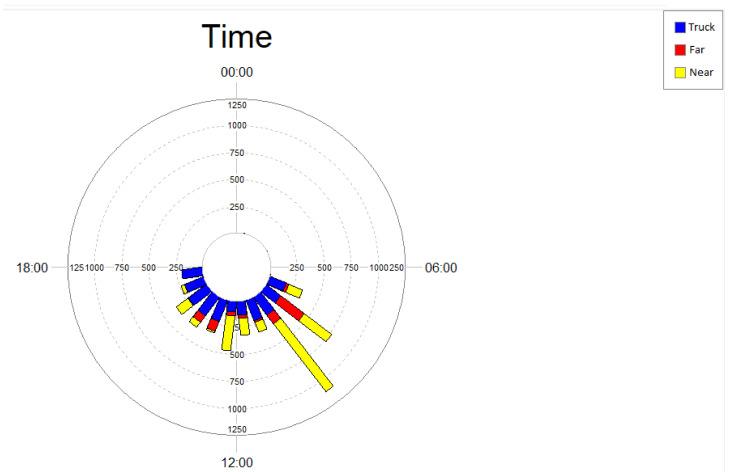
Temporal distribution of *phee*-type calls of black-tufted marmosets (*Callithrix penicillata*) in the area near (yellow) and the area far (red) from the Brucutu Mine, and number of noise events generated by the truck passes detected on the road near the mine (blue), Peti reserve, Minas Gerais, Brazil.

**Table 1 animals-13-00352-t001:** Results of the generalized linear models comparing noise levels (in decibels) and spectral and temporal parameters (duration, minimum frequency, maximum frequency, peak frequency, average power and bandwidth) of *phee* calls emitted by black-tufted ear marmosets (*Callithrix penicillata*) in two different areas, one near and one far from the Brucutu Mine, state of Minas Gerais, Brazil.

Acoustic Parameters *	Estimate ± SE	Z	*p*-Value
Noise (dB) Intercept	−0.3842 ± 0.0023 6.5929 ± 0.0028	−164.00 2278.00	*p* < 0.001 *p* < 0.001
Call Rate Intercept	−1.1722 ± 0.0450 6.0088 ± 0.3902	−26.05 15.40	*p* < 0.001 *p* < 0.001
Duration Intercept	−0.0401 ± 0.0015 6.8665 ± 0.0019	−26.65 3511.30	*p* < 0.001 *p* < 0.001
Minimum Frequency Intercept	0.0513 ± 0.0005 8.6662 ± 0.0007	90.96 −11,642.00	*p* < 0.001 *p* < 0.001
Maximum Frequency Intercept	0.0091 ± 0.0001 1.1150 ± 2.2140	54.16 50,352.00	*p* < 0.001 *p* < 0.001
Peak Frequency Intercept	0.0464 ± 0,0001 1.1040 ± 0.0002	269.20 48,533.00	*p* < 0.001 *p* < 0.001
Average Power Intercept	−1.8754 ± 0.0019 9.8048 ± 0.0020	−966.10 4736.20	*p* < 0.001 *p* < 0.001
Bandwidth 90 Intercept	−0.1114 ± 0.0007 8.4372 ± 0.0009	−153.10 9002.40	*p* < 0.001 *p* < 0.001

* Data for all variables were compared between close area and far area; the close area representing the reference group.

## Data Availability

The data presented in this study are available on request from the corresponding author.

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
