# Peer review of "Acoustic Monitoring of Black-Tufted Marmosets in a Tropical Forest Disturbed by Mining Noise"

_animals, 2023, doi:10.3390/ani13030352_

Round 1
Reviewer 1 Report
The MS investigate the vocalizations emitted by black-tufted marmosets and its impact by noise produced by mining activity. This study aimed to understand whether signal emitted by black-tufted marmosets are affected by mining noise. However, I have following major critiques that need to be answered before I can give a recomendation for this MS.
First, its unclear how the authors deal with the data during statistic analysis. Generalized linear models were used during the analysis of the animal acoustic parameters and noise levels, however, The acoustic landscape were recorded between the periods of 5:00 am and 6:00 pm, whereas, From Line 161-162, we know that for the noise data, the measurements were calculated based on randomly spaced 20-minute intervals between 5:00 am and 6:00 pm. Then how did the author match the two dataset during GLM model analysis.
Secondly , Its not clear to me what's the meaning in table 1, Why two row dates are presented for each parameters.
some further minor comments including:
1. Latitude and longitude need to be added in figure 1.
2. Window size, fft size and temporal and frequency resolution should be given for the spactrogram showed in figure 2.
3. table legend of table 1, Its unclear what's the meaning of the last sentence: areas near and far from Brucutu Mine* state of Minas Gerais, Brazil.
Author Response
The MS investigate the vocalizations emitted by black-tufted marmosets and its impact by noise produced by mining activity. This study aimed to understand whether signal emitted by black-tufted marmosets are affected by mining noise. However, I have following major critiques that need to be answered before I can give a recomendation for this MS. First, its unclear how the authors deal with the data during statistic analysis. Generalized linear models were used during the analysis of the animal acoustic parameters and noise levels, however, The acoustic landscape were recorded between the periods of 5:00 am and 6:00 pm, whereas, From Line 161-162, we know that for the noise data, the measurements were calculated based on randomly spaced 20-minute intervals between 5:00 am and 6:00 pm. Then how did the author match the two dataset during GLM model analysis?
Answer: We matched the animals’ calls (extracted from the records of the acoustic landscape) with the noise decibel levels in the two sampling sites, averaged per hour, in order to have a corresponding noise level (page 5).
Secondly , Its not clear to me what's the meaning in table 1, Why two row dates are presented for each parameters.
Answer: The Intercept (second line in each variable) is related to data distribution, and brings information about the size of the effect. It is standard to include Intercept values in the results of GLMs (page 5).
some further minor comments including:
- Latitude and longitude need to be added in figure 1.
Answer: Thank you, we included latitude and longitude.
- Window size, fft size and temporal and frequency resolution should be given for the spactrogram showed in figure 2.
Answer: Thank you, we included it.
- table legend of table 1, Its unclear what's the meaning of the last sentence: areas near and far from Brucutu Mine* state of Minas Gerais, Brazil.
Answer: We meant the GLMs were run comparing data from the area far with data from the area near the Brucutu Mine*, which is located in the state of Minas Gerais, Brazil. We adjusted the legend, for clariness (page 5).
Thank you for receiving our manuscript, and considering it for review. We appreciate your time and look forward to the referees’ comments in due course.
Reviewer 2 Report
General points to address:
How far are the places from each other? Because on a large spatial scale, calls and responses can differ between populations and may not be a result of a response to the mining activity but rather a result of prior existing call differences.
Sample size: it needs to be discussed as a limitation, that there are only a few areas, and that more areas would have been helpful.
Also, the effect of different seasons might be analyzed.
Minor
Line 35: are or area?
Line 86: 2010?
Line 100: there needs to be more explanation of near and far…is the far place the one where the mining activity cannot be heard or recognized? If so, different words might be used, like “with mining noise” or mining noise absent/present. Further, the distances in miles/km between both places and to the mining activities need to be shown.
See Figure 1: here you use “close”, and in the text use “near”. Please be consistent.
Please discuss whether 1 km is far enough…another place 10 km away may also be relevant for a comparison.
Line 132: this would mean that differences between the “populations” may have already developed?
Line 183: here you use “noisier”. Please be consistent by choosing on term.
Author Response
Reviewer 2
How far are the places from each other? Because on a large spatial scale, calls and responses can differ between populations and may not be a result of a response to the mining activity but rather a result of prior existing call differences.
Answer: The “near” site is approximately 2300 meters away from the “far” site. We inserted this information in the manuscript, on page 4. Due to the short distance separating the two sites, we think it is improbable that the differences we recorded in calls structures are due to spatial distancing, such as what happens in the development of dialects in some species. For example, De la Torre & Snowdon 2009 found variation in vocal structure in pygmy marmosets population distant 30 km away from each other.
Dialects in Pygmy Marmosets? Population Variation in Call Structure. STELLA DE LA TORRE1ANDCHARLES T. SNOWDON. 2009. American Journal of Primatology: 71-333-342.
Sample size: it needs to be discussed as a limitation, that there are only a few areas, and that more areas would have been helpful.
Answer: Thank you; we included this aspect in the discussion, pages 7 to 8.
Also, the effect of different seasons might be analyzed.
Answer: Differently from the birds, which vocalize more during the reproductive season, in marmosets the rate of vocalizations does not vary among seasons. These animals reproduce twice a year, both in the rainy and in the dry season, and the rate of vocalizations is constant along the year (Duarte et al., 2011; Santos et al., 2017). Therefore, we think that an analysis per season is not going to present relevant information.
Minor
Line 35: are or area?
Answer: The correct term is “area”, thank you, we corrected it.
Line 86: 2010?
Answer: Thank you, the correct is 10, and not 2010. Correction made.
Line 100: there needs to be more explanation of near and far…is the far place the one where the mining activity cannot be heard or recognized? If so, different words might be used, like “with mining noise” or mining noise absent/present.
Answer: The mining noise is not completely absent in the far site. It is possible to hear some extreme noises, for example train horns and explosions. This is why we decided to name the sites as “far” and “near”. However, the main source of noise produced by the mine is the truck traffic, and the far site is not disturbed by it. As the noise measurements showed, there is statistical difference between the noise levels in the two sites. We included the noise values (dB) in the results, on the page 5.
Further, the distances in miles/km between both places and to the mining activities need to be shown.
Answer: Thank you, we included the distances between the areas in the methods session, on page 4.
See Figure 1: here you use “close”, and in the text use “near”. Please be consistent.
Answer: Thank you, we corrected it.
Please discuss whether 1 km is far enough…another place 10 km away may also be relevant for a comparison.
Answer: Thank you, we included a discussion about this topic.
Line 132: this would mean that differences between the “populations” may have already developed?
Answer: We meant that, considering the home range size of marmosets, the home range of the two groups we studied did not overlap, thus, the two sampling areas were independent in terms of marmoset vocalizations.
Line 183: here you use “noisier”. Please be consistent by choosing on term.
Answer: Thank you, we corrected it to be consistent.
Thank you for receiving our manuscript, and considering it for review. We appreciate your time and look forward to the referees’ comments in due course.
Round 2
Reviewer 1 Report
After judge the revised MS, it apparently that my former critique was not well addressed. Additionally, their reply showed some draback during their research design.
Such that : For the acoustic landscape recorded between the periods of 5:00 am and 6:00 pm, whereas the noise were recallded based on randomly spaced 20-minute intervals between 5:00 am and 6:00 pm. Then they matched the animals’ calls (extracted from the records of the acoustic landscape) with the noise decibel levels in the two sampling sites, averaged per hour, in order to have a corresponding noise level. Considering the short research period and noise level are fluctuate, using the time scale of 1 hour time span to get a single number and reprsent the noise level may not trustworth. Smaller scale of 10 min or 1 min will be much ideal. Additionally, the noise was measured randomly spaced 20-minute intervals and its high probability that the measured time was during the period when no animals sound was produced, however it was used to represent the period when animal make sound and made the result misleading.
Secondly, The generated noise level was wrong. From the Statistic section, the author mentioned that the noise decibel levels were averaged per hour, This method is problematic since the author should not calculate directively from dB level, They should calculate the sound in power level and then translate in dB level.
Thirdly, as my formmer mentioned the GLM result showed in Table 1, For my understanding, the author can use GLM to analysis the difference of spectral and temporal parameters (duration, minimum frequency, maximum frequency, peak frequency, average power and, bandwidth) of phee calls as a function of location and noise level. i.e. they can bult two-way ANOVA ( site * noise level) full factorial design by including site and noise livel into the model as main factors and building interaction terms into the model. Then why intercept was given for each spectral and temporal parameters.
Last but not the least, the sample size for the statistic should be given which can help for evaluating the roubust of the research.
Author Response
Responde to reviewer:
Point 1: After judge the revised MS, it apparently that my former critique was not well addressed. Additionally, their reply showed some draback during their research design.”
Response: We apologize and include information about the noise analyses in the text.
Pont 2: Such that: For the acoustic landscape recorded between the periods of 5:00 am and 6:00 pm, whereas the noise was recorded based on randomly spaced 20-minute intervals between 5:00 am and 6:00 pm. We then matched the animals’ calls (extracted from the records of the acoustic landscape) with the noise decibel levels in the two sampling sites, averaged per hour, in order to have a corresponding noise level. Considering the short research period and noise level are fluctuate, using the time scale of 1 hour time span to get a single number and represent the noise level may not be trustworthy. A smaller scale of 10 min or 1 min would be ideal. Additionally, the noise was measured randomly spaced 20-minute intervals and its high probability that the measured time was during the period when no animals sound was produced, however, this was used to represent the period when animal make sound and made the result misleading.
Response: Actually, the text was misspelled. We improved it by adding new information. We conducted two types of noise analysis:
1 – To elaborate the Oriana’s figure we quantified the truck traffic by counting all instances of noise generated by passing trucks from 5:00 am to 6:00 pm on the road in front of the site near to the mine. This analysis was done using the soundscapes files recorded by the passive monitoring devices (SongMeter 2). Hourly counts were done by audio and visual identification of the trucks’ broadband pattern in spectrograms using the Raven pro software. We have included this information in the manuscript.
2 – In order to have a more robust measure of the noise levels, we conducted noise level measurements using the 2270B&K frequency analyser. To avoid bias in the measured sound pressure levels we excluded all recordings in which we verified high received levels of animal sounds by using BZ 5503 software. We conducted one measurement per hour from 5:00 am to 6:00 pm, with 20 minutes duration each one, in each sampling point per sampling area. The measurements were done in both areas simultaneously, once every two months during data collection. For each noise measurement we calculated the equivalent continuous level (Leq; time average level of sound), which gives an overall indication of the level of exposure to sound pressure in an environment (Duarte et al., 2011 and 2015).
Considering the comment about the short research period and the possible fluctuation of the noise levels, we respectfully disagree, since our study was conducted along one year, with bimonthly data collection, using four passive acoustics sensors and four sound level meters frequency analyzers (2270 B&K). We conducted several noise measurements with 20 minutes of duration each one in each sampling point per month. The duration of the noise measurements is much higher than the usual duration of noise measurements made by acoustic engineers, since the device (2270B&K) was developed to conduct short duration and punctual measurements, differently from the SongMeters2. In Duarte et al., 2011 we conducted 450 noise level measurements and we perceived that noise pollution is not fluctuating like the animal vocalizations. As you can see in the Oriana’s figure, the passage of trucks, the main noise source of the study area, is not fluctuating like the marmoset vocalizations.
Point 3: “Secondly, The generated noise level was wrong. From the Statistic section, the author mentioned that the noise decibel levels were averaged per hour, This method is problematic since the author should not calculate directively from dB level, They should calculate the sound in power level and then translate in dB level.”
Response: Actually, we calculated the equivalent continuous level from each 20-minute sample, which is a logarithmic average used by acousticians to have an idea of the most frequent noise levels measured in a time interval during a sampling event.
Point 4: “Thirdly, as my formmer mentioned the GLM result showed in Table 1, For my understanding, the author can use GLM to analysis the difference of spectral and temporal parameters (duration, minimum frequency, maximum frequency, peak frequency, average power and, bandwidth) of phee calls as a function of location and noise level. i.e. they can bult two-way ANOVA ( site * noise level) full factorial design by including site and noise livel into the model as main factors and building interaction terms into the model. Then why intercept was given for each spectral and temporal parameters.Last but not the least, the sample size for the statistic should be given which can help for evaluating the roubust of the research.”
Response: We do not totally disagree with the reviewer’s comments, but we feel the analyses we have presented are statistically more robust and appropriate than those suggested by the reviewer. It is, indeed, extremely probable that both statistical approaches will produce the same results (our statistical adviser talks about using the statistically most elegant solution, which we believe ours is). In our scientific opinion a GLM is more statistically appropriate, and we are providing data such as intercepts to be as open as possible about our analyses.
About the comments related to the English language, the original draft was reviewed by two native English speakers (including one of the authors Robert Young who is a British citizen and author of >140 scientific papers). Therefore, we would be grateful if you could highlight to us the specific problems with the English.